# NLRP3 Inflammasome Blocking as a Potential Treatment of Central Insulin Resistance in Early-Stage Alzheimer’s Disease

**DOI:** 10.3390/ijms222111588

**Published:** 2021-10-27

**Authors:** Yulia K. Komleva, Ilia V. Potapenko, Olga L. Lopatina, Yana V. Gorina, Anatoly Chernykh, Elena D. Khilazheva, Alla B. Salmina, Anton N. Shuvaev

**Affiliations:** 1Department of Biochemistry, Medical, Pharmaceutical & Toxicological Chemistry, Krasnoyarsk State Medical University Named after Prof. V.F. Voino-Yasenetsky, 660022 Krasnoyarsk, Russia; ol.lopatina@gmail.com (O.L.L.); yana_20@bk.ru (Y.V.G.); elena.hilazheva@mail.ru (E.D.K.); 2Research Institute of Molecular Medicine and Pathobiochemistry, 660022 Krasnoyarsk, Russia; iluminator@snkip.ru (I.V.P.); chernyh_a@bk.ru (A.C.); allasalmina@mail.ru (A.B.S.); shuvaevanton@hotmail.com (A.N.S.); 3Shared Research Center for Molecular and Cellular Technologies, 660022 Krasnoyarsk, Russia; 4Laboratory of Experimental Brain Cytology, Division of Brain Sciences, Research Center of Neurology, 125367 Moscow, Russia

**Keywords:** Alzheimer’s disease, neuroinflammation, brain insulin resistance, NLRP3 inflammasome, cognitive disorders

## Abstract

Background: Alzheimer’s disease (AD) is a devastating neurodegenerative disorder. In recent years, attention of researchers has increasingly been focused on studying the role of brain insulin resistance (BIR) in the AD pathogenesis. Neuroinflammation makes a significant contribution to the BIR due to the activation of NLRP3 inflammasome. This study was devoted to the understanding of the potential therapeutic roles of the NLRP3 inflammasome in neurodegeneration occurring concomitant with BIR and its contribution to the progression of emotional disorders. Methods: To test the impact of innate immune signaling on the changes induced by Aβ1-42 injection, we analyzed animals carrying a genetic deletion of the Nlrp3 gene. Thus, we studied the role of NLRP3 inflammasomes in health and neurodegeneration in maintaining brain insulin signaling using behavioral, electrophysiological approaches, immunohistochemistry, ELISA and real-time PCR. Results: We revealed that NLRP3 inflammasomes are required for insulin-dependent glucose transport in the brain and memory consolidation. Conclusions NLRP3 knockout protects mice against the development of BIR: Taken together, our data reveal the protective role of Nlrp3 deletion in the regulation of fear memory and the development of Aβ-induced insulin resistance, providing a novel target for the clinical treatment of this disorder.

## 1. Introduction

Alzheimer’s disease (AD) is a devastating neurodegenerative disorder that leads to dementia. It is pathologically characterized by neuronal loss, extracellular amyloid beta (Aβ) deposition and intracellular hyperphosphorylation of tau protein in the brain. Clinically, this is manifested by a decrease in memory and learning, and by abnormal mental, emotional states and behaviors [1]. Over the past decades, extensive efforts have been made to understand the underlying mechanisms of the disease. Scientists have made great strides over the past decades in understanding the pathogenesis of Alzheimer’s disease based on Aβ and tau aggregation, and misfolding, inflammation, oxidative damage, and other factors. However there is currently no cure for AD, although there are treatments available that improve symptoms [2].

In recent years, increasingly attention of researchers has been focused on studying the role of brain insulin resistance formation in Alzheimer’s disease pathogenesis [3]. Therefore, the generally accepted point of view is that insulin resistance in the brain tissue causes the development of cognitive dysfunction and promotes the progression of degenerative changes [4,5,6]. Moreover, it has been shown that brain insulin resistance can develop independently from systemic insulin resistance [7]. A significant contribution to the development of brain insulin resistance is made by neuroinflammation due to the overproduction of proinflammatory cytokines, activation of astroglia and microglia, and disruption of reparative neurogenesis. Chronic, low-grade inflammation is one of the main features observed in diseases associated with neurodegeneration [8] and insulin resistance [9]. Elevated levels of inflammatory markers have been observed in the brain and in the blood of AD patients [9,10].

Pathogen-associated molecular patterns (PAMPs) and endogenous signals, known as danger-associated molecular patterns (DAMPs) released from cells alert the innate immune system and activate several signal transduction pathways through interactions with pattern recognition receptors (PRRs). Both PAMPs and DAMPs directly induce proinflammatory cascades and trigger the formation of the inflammasome, mediating the release of cytokines [11]. Given the fact that Aβ acts as a strong DAMP, it seems that the interval between early accumulation of Aβ and later signs of disease progression, such as tau pathology and brain atrophy, is influenced by innate immune responses [11,12]. One of the canonical pathways of Aβ-induced innate immune response in Alzheimer’s disease is the activation of the NOD-like receptor (NLR) family, a pyrin domain-containing protein, which has been the subject of intense research [8,12]. NLRP3 is a multiprotein complex that consists of an amino-terminal pyrin domain (PYD), a central nucleotide-binding and oligomerization domain (NOD), and a C-terminal leucine-rich repeat (LRR) domain [13]. The NLRP3 inflammasome has an essential role in the deterioration caused by inflammation in Alzheimer’s disease and type 2 diabetes mellitus (T2DM) [14,15].

Recently, much attention has been paid to the study of the multiprotein complex as a possible target molecule in the treatment of many conditions. Our previous studies demonstrated the role of NLRP3 in neurodegeneration and physiological aging, namely the effect on memory and neurogenesis. We previously confirmed that a basal level of NLRP3 expression is necessary for neurogenesis processes, regulating mainly the early stages, i.e., cell proliferation and differentiation. Our data suggest that expression of NLRP3 inflammasomes in neural stem cells and neuroblasts may contribute to the stimulation of adult neurogenesis in physiological conditions, whereas Alzheimer’s type neurodegeneration abolishes stimuli-induced overexpression of NLRP3 within the neurogenic niche [16].

Through Aβ recognition, the NLRP3 inflammasome triggers caspase 1 activation and the processing of cytoplasmic targets, including the proinflammatory cytokines IL-1β and IL-18. Elevated levels of IL-1β may be a contributing factor to insulin resistance in obesity and neurodegeneration [8,14,15]. Insulin receptor substrate 1 (IRS-1) binds insulin to two signaling pathways: the PI3K/Akt and the Ras/ERK pathways [4,9]. One of the most important results of Akt kinase activation by insulin is translocation of the insulin-dependent transporter GLUT4 into the plasma membrane, which leads to stimulation of glucose uptake by cells. This process involves protein kinase C (PKC), one of the targets of Akt kinase. Akt kinase inhibits the activity of GSK3, which leads to the blocking of both negative and positive effects of GSK3. This determines the key role of the IRS-1/PI3K/PKC/GSK3 cascade in realizing the effects of insulin on gene expression, apoptosis, and cell survival. The Ras/ERK pathway modulates cell growth, survival, gene expression and long-term memory (LTM) formation and expression [17,18].

Moreover, brain insulin resistance may therefore constitute a joint pathological feature of metabolic and cognitive dysfunctions [19]. In contrast to the abundance of research results on the mechanisms of cognitive functions dysregulation, the mechanisms for the implementation of complex behavior forms in the insulin-resistant and neurodegenerative brain are practically unexplored. The literature contains only of isolated observations on the mechanisms of disturbance of complex behavior forms and emotional status in this pathology, while the clinical significance of these events cannot be overestimated [20]. Patients suffering from AD have significant memory impairments associated with emotional alterations, which indicates the involvement of the amygdala in this process. Interestingly, there have recently appeared studies on the role of the amygdala in insulin resistance, for example, in relation to eating behavior, as well as on the molecular mechanisms of insulin effects in this brain structure [21]. At the same time, many unspecified aspects remain in the deciphering of insulin resistance mechanisms in chronic neurodegeneration. Given the apparent synergistic adverse effects of neurodegeneration and insulin resistance, it is important to determine how these conditions are related and to explore potential therapies such as blocking the NLRP3 inflammasome. This study was devoted to the understanding of the potential therapeutic roles of the NLRP3 inflammasome in neurodegeneration occurring concomitant with brain insulin resistance and its contribution to the progression of emotional disorders. To test the impact of innate immune signaling on the changes induced by Aβ1-42 injection, we analyzed animals carrying a genetic deletion of the Nlrp3 gene. Thus, we studied the role of NLRP3 inflammasomes in health and neurodegeneration in maintaining brain insulin signaling. In the current research we revealed that NLRP3 inflammasomes are required for insulin-dependent glucose transport in the brain and memory consolidation. However, Nlrp3 knockout protects mice against the development of insulin resistance.

## 2. Results

### 2.1. NLRP3 Knockout Attenuates Learning in a Fear-Conditioning Paradigm in Amyloid beta 1-42-Induced Toxicity

We studied the formation of conditioned fear in the FC test in NLRP3KO mice after injection of soluble Aβ1-42 oligomers to interpret the effect of early changes in the brain. The “learning curve” was examined in mice on the first day of testing (Figure 1A). For the first conditioning day, a significant influence of the factors was revealed separately, as well as the influence of the interaction of these factors when comparing the NLRP3KO and WT groups with the Aβ or PBS injection groups. The was significant influence of the Group factor (F (3.32) = 19.25, *p* < 0.0001); Time (F (2.64) = 55.4, *p* < 0.0001); Subjects (matching) (F (32.64) = 3.179, *p* < 0.0001); influence of interaction of all factors (F (6.64) = 3.982, *p* = 0.0019). To characterize the learning curve (memory acquisition), the freezing time between CS-US1 and CS-US3 was compared. In wild-type animals treated with PBS (WT + PBS), there was a statistically significant difference in the freezing time between the first CS-US1 (39.6 ± 2.49%) and the third CS-US3 (59.32 ± 6.0 percentage) interval (*p* < 0.0001 post-hoc Tukey’s multiple comparisons test), while the percentage of freezing in WT + Aβ mice did not differ. The mice did not show differences in the freezing time in CS-US1 (26.47 ± 33.83 percentage) compared with CS-US3 (33.83 ± 0.46 percentage) (*p* = 0.0801 post-hoc Tukey’s multiple comparisons test). In addition, statistically significant differences were revealed between the groups of wild-type mice injected with PBS and oligomers of beta-amyloid in the first CS-US1 interval (*p* = 0.0190), second CS-US2 (*p* < 0.0001) and in the third CS-US3 interval (*p* < 0.0001 post-hoc Tukey’s multiple comparisons test). However, it was interesting to observe that in NLRP3KO mice, when injected with Aβ1-42 oligomers, a difference in the freezing time in CS-US1 (12.58 ± 0.75 percentage) and CS-US3 (41.44 ± 2.46 percentage) was significant (*p* < 0.0001 post-hoc Tukey’s multiple comparisons test). It should be mentioned that freezing time was statistically different in WT + Aβ mice and NLRP3KO + Aβ mice (*p* = 0.0116 post hoc Tukey’s multiple comparisons test). Thus, it can be said that injection of beta-oligomers for the purpose of inducing neuroinflammatory reactions in the brain of NLRP3KO mice did not lead to changes in the formation of the conditioned fear.

On the second day of testing, when memory retrieval in context (contextual memory) was analyzed, the freezing time was compared between four groups: WT + PBS, WT + Aβ, NLRP3KO + PBS, NLRP3KO + Aβ. There were statistically significant differences at *p* = 0.0062, (Kruskal–Wallis test) between groups. Statistically significant differences were noted between WT + PBS (67.71 ± 2.52 percentage) and WT + Aβ (35.91 ± 3.21 percentage) (*p* = 0.0202, Dunn’s test), as well as between WT + PBS (67.71 ± 2.52 percentage) and NLRP3KO + PBS (34.72 ± 7.99 percentage) (*p* = 0.01, Dunn’s test). Therefore, it was found that the freezing time was reduced in wild-type animals with Aβ injection and was associated with the reduce environment recognition as a potentially dangerous place, and, consequently, impaired memory retrieval processes in a specific context. In the NLRP3KO groups, the freezing time with different operations did not differ, which probably indicates the absence of the effect of the action of amyloid beta oligomers on contextual memory (Figure 1B).

On the third day of testing (cued) in a new environment, when comparing the groups with each other, statistically significant differences at *p* = 0.001 (Kruskal–Wallis test) were revealed. The statistically significant differences were in the groups WT + PBS (73.09 ± 1.92 percentage) and WT + Aβ (45.26 ± 4.3 percentage) (*p* = 0.0005, Dunn’s test), as well as between the WT + PBS groups (73.09 ± 1.92 percentage and NLRP3KO + PBS (45.26 ± 4.3 percentage) (*p* = 0.0005, Dunn’s test). Similar to the observations on the second day, he freezing time was reduced in wild-type animals after introduction of beta-amyloid oligomers due to the absence of conditioned fear formation after the introduction of white noise in a new environment. Freezing time between NLRP3KO + PBS and NLRP3KO + Aβ, as well as on the second contextual day, did not differ, which may indicate no effect of beta-amyloid oligomer action on signal memory (Figure 1C). It is worth noting that the distinction in the learning curve among WT and NLRP3KO mice injected with Aβ1-42 underlines the exacerbating role of the NLRP3 inflammasomes in AD-associated memory decline, which corresponds to a previous report [8].

In the early 1990s it was shown that memory consolidation requires the process of glycogenolysis with the formation of lactate [22]. Considering the evidence identifying astrocytes as the main producers of lactate, a possible role of lactate transfer from astrocytes to neurons in the processes of memory consolidation was suggested [23]. Similarly, lactate has been shown to be required for memory consolidation in other learning tasks in rodents, including spatial working memory [24] and the cocaine preference test (amygdala-related behavior) [25,26,27]. Recently, it was shown that L-lactate produced by astrocytes facilitates the synchronization of the amygdala with the anterior cortex and is involved in decision making [28].

In this regard, we investigated the concentration of the glucose metabolism product, lactate, in the brain amygdala. Thus, we studied the basal lactate level in animals of the control WT group and in NLRP3 knockout mice on the 10th day after Aβ-injection (Figure 2A). Comparison of the groups by two-way ANOVA revealed no significant interaction of the two factors (genotype and operation) (F (1.28) = 0.0024, *p* = 0.9615), as well as no influence of the genotype (F (1.28) = 0.0027, *p* = 0.9590) and operations (F (1.28) = 0.0115, *p* = 0.9155). Multiple comparisons revealed that the level of lactate in the amygdala in wild-type and NLRP3 knockout animals after sham surgery with PBS injection did not differ (23.02 ± 3.68, and 23.4 ± 6.79 nmol/μg protein, respectively, *p* = 0.9999, Sidak’s correction). At the same time, AD modeling by injection of beta-amyloid did not cause significant changes in lactate in NLRP3 knockout mice (23.375 ± 5.34 nmol/μg protein) and in wild-type mice (22.37 ± 8.78 nmol/μg protein) (*p* > 0.9999, Sidak’s correction). Thus, there were no differences in the concentration of lactate in the amygdala of the brain among the studied groups.

Since the activation of the NLRP3 inflammasome triggers the activation of caspase-1 and leads to the maturation of proinflammatory IL-1β and IL-18, the IL-1β evaluation indicates the functional activity of the NLRP3 inflammasome. We next looked at whether Aβ, as a DAMP, led to IL-1β release in NLRP3 knockout mice. IL-1β production has long been associated with both beneficial and harmful effects in neuroinflammation and neurodegeneration [29]. In this study, the concentration of IL-β in the amygdala of the brain was studied in wild-type and NLRP3 knockout mice in a model of neurodegeneration (Figure 2B). Comparison of the groups by two-way ANOVA revealed a statistically significant influence of the interaction of two factors (genotype and operation)—F (1.28) = 5.171, *p* = 0.0308, as well as a statistically significant influence of each factor separately: genotype F (1.28) = 35.99, *p* < 0.0001; operation F (1.28) = 4.99, *p* = 0.0336. When studying the level of IL-1β in all groups of NLRP3 knockout animals, trace amounts of IL-1β were revealed. Multiple comparisons revealed an increase in the IL-1β expression in the brain amygdala in the WT group when simulating Alzheimer’s disease (WT + Aβ) (43.6 ± 8.84 pg/mg protein) compared with sham-operated mice after PBS injection (WT + PBS) (21.52 ± 4.2 pg/mg protein) (*p* = 0.0174, Sidak’s correction). Thus, injection of beta-amyloid caused an increase in the concentration of interleukin 1 beta in the amygdala of wild-type mice, in contrast to the NLRP3 knockout mice where no differences were observed.

It should also be noted that it was confirmed that NLRP3 knockout mice did not express NLRP3, which was manifested by the absence of an immunosignal upon immunostaining, while the expression of NLRP3 was confirmed in wild type animals (Figure 3C).

### 2.2. Alteration of Synaptic Transmission and Insulin-Induced Neuromodulation in the Basolateral Amygdala

#### 2.2.1. Alteration of Synaptic Transmission in Synapses of Amygdala Neurons

In neurons of the basolateral amygdala, a pronounced effect on synaptic transmission was observed after intrahippocampal administration of amyloid-β by changes in electrophysiological parameters. Aβ injection led to an in the fEPSP amplitudes of amygdala neurons in WT mice (0.8 ± 0.1 mV) and NLRP3 KO mice (0.8 ± 0.2 mV) compared to the sham-operated control after PBS injection (*p* < 0.05). The NLRP3 knockout mice after PBS administration had the lowest fEPSP amplitude (0.4 ± 0.1 mV) (Figure 3A–C). Interestingly, the NLRP3 KO + PBS mice had a longer rise time relative to WT + PBS animals (1.8 ± 0.3 ms and 1.0 ± 0.1 ms, respectively) (*p* < 0.05) and to NLRP3 KO + Aβ (1.0 ± 0.1 ms) (*p* < 0.05) (Figure 3A,B,E).

At the same time, a significant difference was found in the coefficient of paired impulses between the amygdala neurons of WT + Aβ and NLRP3 KO+ Aβ (1.1 ± 0.1 and 0.9 ± 0.0, respectively) (Figure 3D). Amyloid Aβ 1-42 led to a decrease in decay time in WT mice (1.3 ± 0.2 ms) compared to WT + PBS (3.3 ± 0.5 ms) (*p* < 0.05) but did not affect NLRP3 KO mice (2.4 ± 0.3 ms) compared to NLRP3 KO + PBS (2.3 ± 0.4 ms) (Figure 3F). Therefore, in the NLRP3 knockout mice, synaptic transmission was impaired. Aβ acted multidirectionally on wild type and knockout mice.

#### 2.2.2. Effect of Insulin Application on Synaptic Transmission in the Brain Amygdala

To assess the effects of insulin on the neurons, we carried out a single acute application of 500 nM insulin solution directly into a chamber with an acute brain section. The electrophysiological activity of neurons was continuously registered while changes were assessed by increasing or decreasing the fEPSP amplitude. The most pronounced changes were observed in the control animals NLRP3 KO + PBS. When 500 nM insulin was added, the fEPSP amplitude dynamics were opposite to those observed in WT + PBS animals (109.2 ± 4.2% and 97.5 ± 3.7%, respectively at point 2 and 115.1 ± 6.2% and 90.3 ± 5.0%, respectively, at point 3) (Figure 4A–C).

In WT and NLRP3 KO animals, Aβ injection disrupted the above-identified pattern of changes in fEPSP amplitude of the amygdala neurons: the amplitudes changed in different directions in time with insignificant deviations from the control values (100%). Moreover, the reliability of the changes in time were not statistically confirmed (Figure 4A–C). Under physiological conditions (WT animals with 1 µL of PBS injected intrahippocampally), the effect of insulin was manifested by a decrease in neuronal excitability. In NLRP3 KO animals in the basolateral amygdala neurons, opposite dynamics were observed, i.e., an increase in excitability with an increase in the fEPSP amplitude. Amyloid-β disrupted this pattern.

### 2.3. NLRP3 Deletion Ameliorates Insulin Resistance in Aβ-Induced Neurodegeneration

The expression of serine phosphorylated insulin receptor substrate (pIRS-Ser) in NLRP3 knockout mice injected with amyloid beta Aβ1-42 was measured. We evaluated the colocalization coefficient using Olympus Fluoview software. We previously studied markers of insulin resistance (GSK3 and PKC) in the context of the development of neurodegeneration in mice after injection of beta-amyloid [30]. However, in the current study, the data were significantly expanded, and other markers were included. The results obtained indicate a statistically significant effect of the interaction of two factors on the expression of pIRS-Ser (F(1.16) = 17.32, *p* = 0.0007, two-way ANOVA). The influences of the operation (F (1.16) = 25.38, *p* = 0.0001) and genotype (F (1.16) = 5.81, *p* = 0.028) were also significant. In the multiple comparison of the overlap coefficients according to Mander’s test, the following results were obtained. In the group WT + PBS the overlap coefficient was low and amounted to 0.24 ± 0.17. When amyloid beta was injected into the brain of wild-type animals, neuronal expression of the insulin receptor substrate phosphorylated by serine was significantly increased (Figure 5A,B) and the overlap coefficient was 0.87 ± 0.15 (*p* < 0.0001, Sidak’s correction). A very interesting finding is that the expression of IRS by mature neurons in animals with an Nlrp3 deletion did not change regardless of the introduction of beta-oligomers. Thus, in NLRP3KO knockout mice, when modeling Alzheimer’s disease by Aβ injection, an overlap coefficient of 0.42 ± 0.18 was revealed, compared with that in the NLRP3 KO + PBS group of 0.36 ± 0.1 (*p* > 0.05, Sidak’s correction). Thus, it can be assumed that deletion of Nlrp3 had a protective role in the development of Alzheimer’s disease or mild cognitive impairment accompanied by insulin resistance due to decreased expression of pIRS-Ser.

We also measured insulin in the brain amygdala to analyze how its expression level changes when modeling Alzheimer’s disease in NLRP3 knockout mice. It is known that type 2 diabetes mellitus increases the risk of developing AD by two to four times [31]. Diabetes mellitus is characterized by hyperinsulinemia, insulin resistance and hyperglycemia. Current evidence suggests that hyperinsulinemia, even without diabetes mellitus, can double the risk of developing Alzheimer’s disease [32]. Therefore, high insulin, in particular, can modulate the risk of AD in several potential ways [33]. In this experiment, the level of insulin in the hippocampus and amygdala of the brain was studied by the method of enzyme-linked immunosorbent assay.

During assessment of the insulin concentration in the amygdala in the control group and NLRP3 knockout mice, no significant differences were found in the insulin expression. Indeed, the influence of the interaction of two factors (genotype and operation) (F (1.28) = 3.314, *p* = 0.0794, two-factor ANOVA), as well as the genotype factor (F (1.28) = 2.495, *p* = 0.1255) and operations (F (1.28) = 0.4199, *p* = 0.5223) (Figure 5C) on insulin level were not significant. It should be noted that when beta-amyloid was administered to control mice (WT + Aβ), there was a tendency towards an increase in insulin (0.345 ± 0.075 ng/mg protein) compared to sham-operated mice (WT + PBS) (0.185 ± 0.051 ng/mg protein) (*p* = 0.32, Sidak’s correction). Injection of beta-amyloid in NLRP3 knockout mice did not affect the insulin concentration in the brain (0.125 ± 0.082 ng/mg protein) compared to PBS injection (0.2 ± 0.082 ng/mg protein) (*p* = 0.84, Sidak’s correction).

#### Study of the Features of Glycogen Synthase Kinase-3 (GSK3β) and PKC Expression in Neuronal Cells

A study was carried out in the control group with sham surgery (WT + PBS), in simulating Alzheimer’s disease in wild-type animals (WT + Aβ), in NLRP3 knockout animals with sham surgery (NLRP3 KO + PBS) and in NLRP3 knockout mice with the introduction of beta amyloid (NLRP3 KO + Aβ).

During the GSK3β evaluation, a statistically significant effect of interaction was revealed (Interaction F (1.24) = 11.05, *p* = 0.0028; operation Aβ or PBS F (1.24) = 32.29, *p* < 0.0001; genotype F (1.24) = 25.39, *p* < 0.0001, two-way ANOVA, post-hoc Sidak’s multiple comparisons test). In particular, we recorded an increase in GSK3 expression in mature neurons in the WT + Aβ (0.63 ± 0.2) group compared to the control WT + PBS (0.21 ± 0.03) (*p* < 0.0001) (Figure 6A,B). There was no significant increase in glycogen synthase kinase-3 expression after intrahippocampal administration of amyloid beta to the NLRP3 knockout animals (0.24 ± 0.1) compared to the group of sham-operated animals NLRP3 KO + PBS (0.13 ± 0.1) (*p* > 0.05).

We analyzed the expression of PKC in the neurons of the studied groups (Figure 7A,B). The highest expression in neurons was observed in the NLRP3 knockout mice with the sham operation (NLRP3 KO + PBS), where the colocalization coefficient was 0.88 ± 0.12, which did not significantly differ from the same group of wild-type animals (WT+ PBS) (0.76 ± 0.2) (*p* = 0.4814). At the same time, we observed a significant decrease in PKC expression in wild-type animals after modelling neurodegeneration (0.41 ± 0.1) compared to NLRP3 KO knockout animals with amyloid beta injection (0.69 ± 0.17) (*p* = 0.0171) and compared to WT + PBS (*p* = 0.0025). Statistical analysis using the two-way ANOVA, post hoc Sidak’s multiple comparisons test showed a statistically significant effect of the Operation F (1.23) = 20.49, *p* = 0.0002 and Genotype F(1.23) = 11.24, *p* = 0.0028). Therefore, Aβ affected GSK3β and PKC expression in control group, but not NLRP3 knockout group.

### 2.4. NLRP3 Inflammasome Is Essential for Insulin-Dependent Glucose Transport in the Amygdala

Insulin resistance is defined as a process in which normal or elevated insulin levels produce a reduced biological response characterized by impaired sensitivity to insulin-mediated glucose disposal [34]. As described in the introduction, neuroinflammation is a common feature of many neurodegenerative diseases, including Alzheimer’s disease. Therefore, we decided to evaluate the expression of GLUT4 and IRAP markers as well as insulin mRNA in animals that do not express NLRP3 inflammasomes to assess the contribution of neuroinflammation to the development of insulin resistance. Insulin plays important regulatory roles in brain, where it interacts with the insulin receptor in neurons located in various brain areas [35]. In NLRP3 KO mice, a statistically significantly higher expression of insulin mRNA in the amygdala (10.44 ± 1.996) was found in comparison with wild-type mice (4.435 ± 0.843) (*p* = 0.0145, Mann-Whitney U test) (Figure 8A).

Since the translocation of GLUT4 into the cell membrane occurs by IRAP activity, the absence of IRAP expression can lead to a decrease in GLUT4 expression [36]. In addition, in recent years IRAP has been considered as one of the markers of insulin resistance in AD, but its mechanism of action has been the subject of active research by scientists [37,38,39]. Colocalization analysis of GLUT4/IRAP molecules was performed in basolateral amygdala. The data obtained indicate a decrease in the number of cells that express GLUT4/IRAP in NLRP3KO knockout mice (8.5 ± 1.75) compared with wild-type control animals (29.75 ± 3.18) (*p* < 0.0001, Mann-Whitney U test) (Figure 8B–D).

In mice not expressing NLRP3, there was a statistically significant decrease in the number of mature neurons expressing insulin-dependent aminopeptidase (2.05 ± 0.30) compared with control mice (28.25 ± 4.10) (*p* < 0.0001, Mann-Whitney U test) (Figure 9A–C). In GFAP-positive astrocytes, a significant decrease in the expression of IRAP was noted in the group of NLRP3KO mice (2.64 ± 0.35 cells) compared with mice in the control group (13.6 ± 2.16) (*p* = 0.0003, Mann-Whitney U test) (Figure 9D). Thus, study of the expression features of the insulin-dependent aminopeptidase IRAP marker in the brain of NLRP3KO mice demonstrated a significant decrease in this marker in glial and neuronal cells as compared to wild-type mice. A decrease in the expression of the insulin-dependent glucose transporter GLUT4 in the basolateral amygdala was also noted. At the same time, literature data indicate that IRAP plays a significant role in postnatal brain development and normal functioning of the hippocampus in adulthood [40,41].

#### Expression of MAPK (ERK1/2) in NLRP3 Knockout Mice

It is known that the mechanism of insulin signal transduction through insulin receptors is realized through two main signaling pathways, namely, the phosphatidylinositol 3-kinase pathway (PI3K/Akt pathway) and the mitogen-activated protein kinase pathway (MAPK pathway). Moreover, the MAPK pathway primarily regulates the nonmetabolic effects of insulin, such as cell growth, proliferation, differentiation, and survival [18,42]. ERK kinases can phosphorylate and activate various cytosolic proteins, phospholipase A2, cytoskeletal proteins, and tyrosine kinases. The ability of insulin to activate MAPK kinase expands the spectrum of its biological effects and ensures the interaction of the insulin system with other signaling systems [43]. Therefore, we studied the expression of MAPK (ERK1/2) in NLRP3 knockout mice. We did not find statistically significant differences in the expression of this marker in NeuN-positive cells (Figure 10A). In both groups, most cells in the field of view expressed MAPK (ERK1/2). In the group of NLRP3 knockout mice, the number of cells in the field of view expressed ERK1/2 15.74 ± 2.69, in control mice (12.25 ± 1.13) (*p* = 0.26). However, it should be noted that the control mice had a higher expression intensity of MAPK (ERK1/2) extracellularly (146.9 ± 15.02) compared with NLRP3 knockout mice (129.15 ± 11.95) (*p* = 0.0308) (Figure 10B).

Previously it was demonstrated that DAMPs and PAMPs induced NLRP3 expression through ERK1/2 and JNK1/2, and induced proIL-1β expression through p38 [44]. Therefore, a slight decrease in the intensity of the ERK1/2 marker immunofluorescence was observed upon NLRP3 gene deletion.

## 3. Discussion

To test the impact of innate immune signaling on the changes induced by Aβ1-42 injection, we analyzed animals carrying a genetic deletion of the *Nlrp3* gene, thus lacking NLRP3 inflammasome expression, a central participant of peripheral and cerebral innate immunity. To maintain homeostasis, the immune system must detect and repair damage to sterile tissue and provide clearance from toxic metabolites such as amyloid-beta oligomers. The NLRP3 inflammasome coordinates these processes to restore tissue damage and induce repair. However, dysregulation of the NLRP3 inflammasome activity causes many conditions, including Alzheimer’s disease [45].

### Dual Role NLRP3 the Inflammasome in Health and Disease

In our research, we studied the role of NLRP3 inflammasomes in health and neurodegeneration in maintaining brain insulin signaling, and a dual role of NLRP3 has been described. When released under homeostatic conditions in a healthy brain, NLRP3 and IL-1β activate cells to produce lactate. This is important for long-term memory consolidation. Similar to IL-1β action, as described by Goshen and colleagues, physiological levels of NLRP3 promote memory formation [12], and NLRP3 inflammasomes are required for insulin-dependent glucose transport in the brain. At the same time, NLRP3 knockout ameliorates insulin resistance during the development of amyloid-induced neurodegeneration. This is reflected in improved cognitive function and learning in the fear conditioning test.

There are scattered data in the literature on the role of inflammasomes and proinflammatory interleukins during Alzheimer’s disease. Thus, according to some data, NLRP3 knockout mice crossed with APPswe/PS1ΔE9(APP/PS1) mice (AD genetic model) are completely protected from amyloid-induced neurodegeneration [46].This is due to the lack of production of mature IL1β or IL18. However, there are studies in which the opposite results were obtained. In IL18KO/APP/PS1 mice, deaths were observed at the age of 2 months. IL-knockout mice crossed with transgenic AD mice had a lower threshold for chemically induced seizures and a selective increase in gene expression associated with increased neuronal activity. These mice were found to have increased expression of synaptic protein excitability, dendrite density, and basal excitatory synaptic transmission, all of which contribute to seizure activity. IL18KO/APP/PS1 mice exhibited increased expression of excitatory but not inhibitory synaptic proteins [47].

Our own research suggests that inflammasomes are likely to play different roles in different physiological and pathological events. We revealed that freezing behavior was modified by NLRP3 deletion. Wild-type mice with a basal NLRP3 expression showed normal behavior in the fear-conditioning test and preserved memory acquisition ability during training, which is consistent with electrophysiological data, while hippocampal function was also impaired in mice with NLRP3 deletion [48].

In the current study the role NLRP3 knockout in learning the fear-conditioning paradigm after injection of beta oligomers was experimentally confirmed. Thus, it was shown that injections of beta-amyloid did not have a statistically significant effect on mice with knockout of the Nlrp3 gene. In these mice, a typical learning curve was observed, which indicates the preservation of the acquiring memory process and the absence of a negative effect of beta-oligomers on the hippocampus. In addition, here was no negative effect on the memory retrieval on the contextual and signal days. In previous studies, the NLRP3 molecule was proposed as a target molecule for AD therapy; however, the mechanism of the protective action remained completely unexplored [45,49]. Taken together, our data revealed the protective role of Nlrp3 deletion in the regulation of fear memory and the development of Aβ-induced neurodegeneration, providing a novel target for the clinical treatment of this disorder.

Aβ has a pronounced effect on synaptic transmissions in amygdala neurons, both at the pre- and post synapse. In classical works related to inflammation, synaptic transmission modulation under the influence of a pathogenic factor was shown. Results showed an increase in the expression of ligand-dependent receptors (most often AMPA receptors) on the postsynaptic membrane of neurons. This process also takes place in neuroinflammation, in particular, in Alzheimer’s disease [50].These reports correlate with our data and explain the increase in fEPSP amplitudes in the basolateral zone of the amygdala upon administration of Aβ to the animals. Aβ itself cannot have a direct effect on the neurons of the amygdala since it was introduced into the hippocampus. It seems that it induces a cascade of reactions with the release of inflammatory mediators from microglia, such TNFα, interleukins (IL-1β, IL6), and free radicals [51]. This Aβ-mediated cascade of signals extends to neighboring areas of the brain and, in particular, to the amygdala.

In WT and NLP3 knockout animals, the action of amyloid-β is more likely associated with an increase in the expression of AMPA receptors on the postsynaptic membrane and an increase in the excitability of amygdala neurons. Previously it was shown, that hyperexcitability of the basolateral nucleus of the amygdala is associated with increased anxiety and often occurs in parallel with various neurodevelopmental, neurodegenerative, and neuropsychiatric disorders [52,53]. However, an increase or decrease in the expression of AMPA receptors alone cannot explain the entire complex of pathological changes in neurons, which occurs in NLRP3 knockout mice and under the Aβ action. Thus, the increase in the fEPSP amplitude directly indicates the strength of excitatory synaptic transmission, i.e., how quickly the presynapse releases the neurotransmitter and binds to the ligand-dependent channel [54]. According to our data, this is the most sensitive indicator of impaired synaptic transmission in NLRP3 knockout mice. A significant slowdown in the rise time of fEPSP in the basolateral zone of the amygdala was revealed in NLRP3 knockout mice after sham operation.

The decrease in decay time in the WT group caused by Aβ injection, is probably associated with a decrease in cell size. In NLRP3 knockout mice, the Aβ action was associated with a decrease in the release of a neurotransmitter from the presynapse; however, an acceleration of the temporal parameters of this process and, probably, increased expression of AMPA receptors on the postsynaptic membrane had a pronounced excitatory effect in the form of an increase in fEPSP amplitudes. In general, these data indicate a greater sensitivity of amygdala neurons to the Aβ damaging effect in NLRP3 knockout mice. Thus, in NLRP3 knockout mice we observed a decrease in excitability and acceleration of neurotransmitter release to normal values in the amygdala neurons of animals with experimental AD, both at the pre- and postsynaptic levels.

The effect of insulin in the form of a constant decrease in the fEPSP amplitude was revealed and statistically proven in neurons of the basolateral amygdala in WT animals after PBS injection. However, a cascade of inflammatory reactions spreading from the Aβ-affected hippocampus to the amygdala caused a disturbance in the dynamics of the fEPSP amplitudes in the presence of 500 nM insulin. An interesting fact is that NLRP3 knockout mice had an increase in the fEPSP amplitude in the presence of 500 nM insulin. To explain this fact, it is necessary to accept the revealed decrease in the fEPSP amplitude of amygdala neurons in WT animals in response to 500 nM insulin as an abortive long-term depression, where the cascade of AMPA receptors internalization could not lead to the consolidation of the inhibitory effect, i.e., the incorporation of these receptors into endosomes. In this case, it is appropriate to compare the initiation of long-term plasticity at the synapses of the basolateral amygdala in WT mice injected with PBS with other experimental groups. It has been shown that IL-1β plays an important role in the initiation of long-term plasticity, and when blocking this inducer of inflammation, LTP in Schaffer’s collaterals is disrupted in vitro [55]. Therefore, it can be assumed that the absence of a normal background immune response in NLRP3 KO mice in the form of an IL-1β-mediated signaling cascade disrupts insulin-dependent plasticity and leads to effects opposite to those found in WT animals. The observed Aβ effect on insulin-mediated dynamics of the fEPSP amplitudes also correlates with the data described in the literature and fits into the theory of insulin resistance in neurodegenerative diseases such as Alzheimer’s disease [12,19,56,57]. As noted above, insulin, via insulin receptors on the neuron surface, induces clathrin-mediated intralization of AMPA receptors, which leads to a long-term decrease in amplitude [58]. Therefore, a dysfunction of insulin receptors should lead to a decrease in the severity or complete disappearance of the suppressor effect of insulin on fEPSP amplitude due to initiation alteration.

In other words, a basal level of NLRP3 inflammasome is required for successful memory consolidation, i.e., expression of the NLRP3 inflammasome as in the control group, which is involved in the processes of memorization. To further interpret these results the insulin downstream cascade was studied. For the first time, it was shown that Nlrp3 knockout protects mice from the pathological effects of beta-amyloid oligomers, which was manifested by the unchanged level of expression of IRS1-Ser compared to the control. Thus, after injection of amyloid beta into the hippocampus in mice, we observed a decrease in insulin signaling in the brain, primarily due to the expression of dysfunctional IRS-1. This likely reflects Aβ-induced glial secretion of proinflammatory cytokines such as IL1β. As described earlier in studies, microglial IL-1, IL-6, and TNF-α activate serine kinases IRS-1, including ERK2, via neuronal receptors [59]. Increased neural IRS-1 pS is significant in the cerebral cortex and hippocampal formation in AD and, apparently, is the main cause of IRS-1 dysfunction in AD [60,61]. Since this phosphorylation inhibits the transmission of insulin-induced receptor activation to downstream molecules, it becomes clear that this is a common cause of insulin resistance in peripheral tissues and in the brain [62]. As expected in our study, levels of these candidate biomarkers were negatively associated with cognitive ability. Nevertheless, in the Nlrp3 knockout genotype after injection of beta-amyloid, these changes in the expression of insulin signaling markers were not observed.

Studies have shown that increased phosphorylation of IRS1 at serine leads to the inability to transmit signals to secondary messengers such as PI3K. This affects other pathological processes in the brain tissue, including tau phosphorylation and neuroinflammation. PI3K/Akt signaling can mediate several downstream pathways, including the Wnt/β-catenin pathway, mTOR signaling, and regulation of GSK3β activity [63,64]. Thus, an increase in the expression of an aberrantly phosphorylated insulin receptor substrate in neurons determines the formation of local insulin resistance during experimental neurodegeneration.

In this study, it was found that injection of beta-oligomers led to overexpression of GSK3β in the amygdala. The role of NLRP3 inflammasomes in the formation of local insulin resistance in the modeling of Alzheimer’s disease is described for the first time. NLRP3 knockout animals have a weak GSK3β fluorescence signal. Injection of beta oligomers in these animals did not increase GSK3β expression. These results are supported by recent studies that insulin signaling was impaired, including through overexpression of GSK3β, in the brains of mice with type 2 diabetes mellitus with Alzheimer’s disease, characterized by resistance to insulin [65]. Reduced level of PKC in amygdala neurons of amyloid-beta injected mice may at least partially underlie failure of fear learning. Such a relationship was confirmed in amyloid-bets injected NLRP3 knockout mice, which showed neither decreased level of PKC nor impairments in fear learning in comparison with sham operated mice [66].

In the current study, there were no statistically significant differences in lactate in the amygdala among the study groups. Earlier, we described an increase in hippocampal lactate in wild-type animals after modeling Alzheimer’s disease, which was associated with cognitive deficits. It is known that an increase in the hippocampal lactate level is observed with the onset of cognitive deficit in mice with Alzheimer’s disease at the age of 12 months [67]. Thus, despite the similar lactate levels, it is difficult to draw conclusions about normal glucose metabolism in the amygdala of the brain of different groups, since the expression of other markers was impaired.

In this study, insulin concentration was evaluated, since study of the origin of insulin in the central nervous system remains most relevant in modern neuroscience and endocrinology [3]. There were no statistically significant differences in the concentration of insulin in the amygdala in the studied groups with AD modeling. However, we found an increase in insulin mRNA in the amygdala of NLRP3 knockout mice. Previously it was shown that obese and AD patients have CSF insulin concentrations lower than in control subjects, suggesting a reduction in both insulin transport across the BBB and hormone sensitivity, which could be associated with increased proinflammatory cytokines and molecules [6]. In general, GLUT4 and insulin protein and insulin mRNA showed similar differences in abundance between brain areas. However, in some locations, differences were observed between relative abundance of mRNA and protein indicating post transcriptional regulation [35]. Thus, the increase in mRNA and the difference with the protein expressed in brain tissue can be explained by post transcriptional modifications in knockout mice, as well as by different levels of peripheral insulin transport across the BBB.

Earlier in our research, we described the effect of deletion of the NLRP3 gene on memory consolidation and processes of neurogenesis [16,48]. Therefore, based on the data obtained on the role of the IRS-1/PI3K/PKC/GSK3 pathway in modeling neurodegeneration in NLRP3 knockout animals, we further studied the effect of gene deletion on insulin-dependent transport of glucose and ERK kinases that control the expression of genes and transcription factors involved in the processes of neurogenesis and memory.

We hypothesized that the expression and deletion of NLRP3 inflammasome would influence the insulin dependent glucose transport and insulin sensitivity. The results obtained indicate a decrease in the expression of IRAP in the neurons and astrocytes in amygdala in NLRP3 knockout mice that is consistent with data proving the effectiveness of administration of insulin-dependent aminopeptidase inhibitors as “cognitive enhancers”. However, there is evidence that decreased IRAP expression in the hypothalamus is associated with the development of schizophrenia. It has also been suggested that the hypothalamic expression of IRAP may be closely related to the neuropeptide system, since IRAP is a vasopressin and oxytocin-degrading enzyme [68]. This phenomenon is finding additional support after postmortem brain studies of patients with mood disorders. Previously, it was shown that the cell density of IRAP-expressing neurons in the paraventricular nucleus increases in persons with depression [69].

A decrease in GLUT4 expression in the BLA of NLRP3 knockout mice was also demonstrated. It is known that the level of GLUT4 in the brain can depend on the levels of insulin. In addition, in rodents with insulin deficiency and manifestations of diabetes mellitus, a decrease in GLUT4 was observed. Interestingly, GLUT4 colocalizes with the insulin receptor in glucose-sensitive neurons [70]. Thus, the decreased level of the insulin-dependent glucose transporter in NLRP3 knockout mice may indirectly indicate insulin reduction. This is because insulin stimulates the expression of the GLUT4 gene and its transport from the cytoplasm to the plasma membrane, thereby modulating the absorption and utilization of glucose. Therefore, the insulin signaling pathway plays a key role in the regulation of transmembrane glucose transport [71].

As described above, memory consolidation requires suprabasal insulin levels, that are mediated through GLUT4. This explains the impaired memory acquisition on the first day of testing in FC. Moreover decreased GLUT4 expression can hinder the process of memory consolidation [72]. Decreases in the intensity of the ERK1/2 marker immunofluorescence were observed upon NLRP3 gene deletion. This data is consistent with the data of Merlo et al. [73] who suggested that NMDA receptor-dependent activation of different pools of amygdalar ERK1/2 may be required for reconsolidation and extinction of fear conditioning. Similar findings were obtained in amygdala-dependent learning tasks where the increase in ERK1/2 activation was NMDA receptor-dependent [40]. ERK1/2 signaling in this region appears to modulate extinction consolidation and retrieval [17,74].

## 4. Materials and Methods

### 4.1. Experimental Animals

The experiments were carried out using male C57BL/6 mice, 3–4 months old, weighing 25–30 g. An age-matched male B6.129S6-Nlrp3tm1Bhk/JJ line was used with Nlrp3 gene knockout: NLRP3 KO, created at The Jackson Laboratory (USA) on the genetic background C57BL/6.

The mice (5–6 males per cage) were kept in individually ventilated cages at a temperature of 21–22 °C with free access to water and food and a regular light cycle of 12 h day/12 h night. All manipulations were performed during the light phase of the day. During the experiment, every effort was made to minimize the suffering of animals and reduce their number according to the principles of working with animals (Principle 3 “R”). Research on animals was carried out in accordance with the principles of humanity set forth in in the European Community Directive (2010/63/EC) and with the permission of the Bioethical Commission of Professor V.F. Voino-Yasenetsky Krasnoyarsk State Medical University, Krasnoyarsk, Russia.

### 4.2. Experimental Design and Groups of Animals

The study consisted of two blocks:

I block: Investigation of the NLRP3 role in synaptic transmission, memory formation, glucose metabolism and insulin signaling in modeling Aβ-induced neurodegeneration.

Animals were divided into four groups.

WT + PBS group—C57BL/6 mice with the injection of phosphate buffered saline (PBS) into the CA1 zone of the hippocampus (*n* = 15).WT + Aβ group—C57BL/6 mice with the injection of Aβ1-42 into the CA1 zone (*n* = 16).NLRP3 KO + PBS group—NLRP3-knockout mice with the PBS injection into the CA1 zone of the hippocampus (*n* = 20).NLRP3 KO + Aβ group—NLRP3-knockout animals with the injection of Aβ1-42 into the CA1 (*n* = 20).

II block: Evaluation of the NLRP3 expression effect in the brain tissue on insulin-regulated transport of the glucose transporter.

Animals were divided into two groups.

WT group—C57BL/6 wild-type mice (*n* = 15)—control group.NLRP3 KO group—NLRP3 knockout mice (*n* = 20)—experimental group.

The experimental design and timeline of this part is shown in Figure 11.

### 4.3. Stereotaxic Surgery

To model Aβ-induced neurodegeneration, a stereotaxic protocol was applied and soluble oligomeric forms of amyloid Aβ1-42 were injected into the hippocampus of mice according to a previously described procedure [16]. Briefly, stereotactic surgery was performed under general anesthesia with sodium pentobarbital (60 mg/kg) and supplemented throughout the surgery as required. After reaching the required level of anesthesia, mice were placed in a stereotaxic frame with an adapter for mice (Neurostar, Germany). The skin overlaying the skull was retracted to drill holes, and injections were made using a Hamilton microsyringe, which was slowly lowered to the place. Beta-amyloid Aβ1-42 (Sigma-Aldrich, USA) was dissolved in sterile 0.1 M PBS (pH 7.4) and a solution was prepared at a concentration of 50 uM. To obtain soluble forms of beta-amyloid, Aβ1-42 was aggregated by incubation at 37 °C for 7 days prior to administration. The obtained Aβ1-42 were injected intrahippocampally into the CA1 zone along the stereotaxic coordinates ML ± 1.3 mm, in the AP-2.0 mm, DV-1.9 mm on both sides with a microsyringe (Hamilton). The mice of the experimental groups were injected with 1 μL of Aβ1-42. For the animals of the control group, 1 μL of phosphate-buffered saline (PBS) was injected to exclude the effect of the stress of the operation. Aβ1-42 were injected over a 5-min period (0.2 μL/min), and the needle was left in place for another 5 min after injection [75]. The animals were taken to the individual cages after the surgery [76].

### 4.4. Behavioral Testing

#### Contextual and Cued Fear Conditioning

A fear conditioning test was carried out according to the standard technique [77]. The test was performed for three consecutive days. The first day consisted of creating conditions for freezing (conditioning test), the second day the context test, and the third day a cued test, associated with the stimulus of conditioned freezing. The test device was an acrylic square chamber (size 33 × 25 × 28 cm), which was placed in a soundproof chamber (size 170 × 210 × 200 cm) to minimize external noise during testing. A backlight (100 lux LEDs) and a speaker (conditioned stimulus (CS) was mounted above the test chamber. The grid floor was connected to an electrical generator to provide an electrical signal (unconditioned stimulus (US) (Ugo Basile, Italy).

To create conditions for freezing during the first day, the mice were placed in a chamber. The mouse could move freely and explore the chamber for 120 s. After that, an auditory signal (white noise, 55 dB) was presented as a conditioned stimulus (CS) for 30 s, and a 0.3 mA foot shock as an unconditioned stimulus (US) was delivered continuously during the last 2 s of the white noise. To establish the association, the presentation of the CS-US paring was repeated three times per session at 120, 240, and 360 s after the beginning of the conditioning. The mice remained in the chamber for 90 s after the final foot shock and then were replaced into the home cage. The total test duration was 480 s on the first day. The analysis of the obtained results was carried out for each allocated time interval (from 0 to 120 s, from 120 to 240, from 240 to 360 and from 360 to 480 s) to track the dynamics of acquisition of the association of a conditioned stimulus with an unconditioned stimulus.

The context test (second day of testing) was performed 24 h after the first test, when conditions were created to freeze in the same chamber for 300 s in the absence of any stimuli (without the supply of white noise (CS) and electrical signal 0.3 mA (US)).

A cued test (the third day of testing) was carried out the next day after the context test. This test ran for 360 s in a new context environment. The test chamber differed from the chamber used on the first and second days in terms of wall color, floor structure (no grid), and 30 lux illumination, providing a new context that was not related to the chamber in which the freezing conditions were created. The test consisted of a 180-s period of mice exploring a new environment to assess nonspecific contextual fear, followed by a 180-s conditioned stimulus (white noise (CS) without electrical current (US)) to assess acquired fear. The percentage of the duration of the freezing reaction in each time interval was determined as an indicator of fear memory, since freezing, as the most common element in behavior caused by severe pain or fear, is often used for quantitative characterization in the FC test [77,78]. The entire testing process was recorded using the ANY MAZE animal video analysis system (Behavior Tracking Software, Stoelting, Chicago, IL, USA).

### 4.5. Electrophysiology

#### 4.5.1. Acute Slice Preparation

Once deep anesthesia was achieved, the mouse was decapitated and the brain was extracted. Then, the brain was immediately placed into ice-cold Ringer’s solution, perfused by 95% О_2_ + 5% СО_2_, for 1 min. Coronary acute slices 350 μm thick were obtained using a Thermo Scientific MicromHM650V vibratome. The sections were cut in Ringer’s solution contained in mM (234 sucrose, 26 NaHCO_3_, 2.5 KCl, 1.25, NaH_2_PO_4_, 11 glucose, 10 MgSO_4_, and 0.5 CaCl_2_) at a temperature of 4 °C and constant perfusion with 95% О_2_ + 5% СО_2_ [79].

#### 4.5.2. fEPSP Recording

Acute slices were put in a chamber perfused with continuously oxygenated (95% О_2_ + 5% СО_2_) extracellular solution containing 125 mM NaCl, 2.5 mM KCl, 2 mM CaCl_2_, 1 mM MgCl_2_, 1.25 mM NaH_2_PO_4_, 26 mM NaHCO_3_, and 10 mM glucose at pH 7.2 at a rate of ~2 mL/min. Slices were visualized with the help of microscope (Olympus BX51WI). To assess the total synaptic activity of neurons, field Excitatory Postsynaptic Potentials (fEPSP) were recorded from basolateral nuclei of the amygdala by using 5–10 МΩ borosilicate glass electrode filled with extracellular solution. Neuronal activity was elicited by electrical stimulation (0.1 ms duration, 0.33 Hz) using a 5 МΩ platinum iridium electrode. Data registration (3 kHz filter) and conversion into digital format were carried out using an HEKA EPC10 amplifier. In acquired the fEPSP, amplitude, rise time, decay time and PPF (paired-pulse facilitation) ratio were measured. The fEPSP amplitude was measured as 50% of the maximal amplitude. The PPF was recorded with 50 ms interval between first and second fEPSPs and measured as a ratio 2nd fEPSP divided to 1st fEPSP [16].

To assess the effect of insulin on the neurons of the studied areas, the control was recorded for 2.5 min to stabilize the amplitude followed by an acute application of insulin (500 nM) and further recording of fEPSP for 5 min. All amplitudes were averaged during 30 s and normalized to the control data (before insulin application).

### 4.6. Immunohistochemistry and Confocal Microscopy

#### 4.6.1. Immunohistochemistry Procedure

After overdose, anesthetized mice were transcardially perfused with 4% PFA with 0.1M phosphate-buffered saline (Sigma). The brains were removed and post fixed in 4% PFA at 4 °C overnight and then immersed for 48 h at 4 °C in 20% sucrose dissolved in phosphate buffered saline (PBS, pH 7.4) containing 0,01% sodium azide (Merck KGaA, Darmstadt, Germany). Sections were cut 50 μm thick using a vibratome (Thermo Scientific, Waltham, Massachusetts, USA) in the sagittal direction using a stereotaxic atlas to determine the localization of the basolateral amygdala (BLM).

Free floating brain slices were blocked in PBS containing 10% normal goat serum (Sigma), 2% bovine serum albumin (BSA) (Sigma), 1% Triton X-100 (Sigma) and 0.1% sodium azide (Sigma), within 1 h at room temperature.

In the first block of studies, immunohistochemical studies were carried out using the following antibodies and their combinations: anti-IRS1 (phosphor S312) (Abcam, ab66154, rabbit polyclonal) 1:1000 and anti-NeuN (Merk, ABN90, Guinea Pig polyclonal) 1:1000 to assess the expression of the substrate of the insulin receptor phosphorylated by serine, and anti-GSK3β (Abcam, ab69739, rabbit polyclonal) 1:1000, anti-PKC (Abcam, ab23513, rabbit polyclonal) 1:1000.

In the second block of studies, the following antibodies and their combinations were used: anti-GLUT4 (Abcam, ab654, rabbit monoclonal) 1:1000 and anti-GFAP (Abcam, ab4674, Chicken polyclonal) 1: 1000 to assess the expression of an insulin-dependent transporter glucose GLUT4 on GFAP-positive astrocytes; anti-GLUT4 (Abcam, ab654, rabbit monoclonal) 1: 1000 and anti-NeuN (Merk, ABN90, Guinea Pig polyclonal) 1:1000 to assess GLUT4 expression on NeuN-positive mature neurons; anti-GLUT4 (Abcam, ab654, rabbit monoclonal) 1: 1000 and anti-IRAP (Santa Cruz, sc-8481, goat monoclonal) 1:1000 to assess colocalization of insulin-dependent glucose transporter GLUT4 and insulin-dependent aminopeptidase IRAP, anti-MAPK (Abcam, ab201015, rabbit monoclonal) 1:1000 to assess MAPK(ERK1/2) expression on NeuN-positive mature neurons. The day after antibody incubation, sections were washed in PBS and then incubated with Alexa-conjugated secondary antibodies 1:1000 (Invitrogen) for 2 h at room temperature. After washing, sections were mounted on glass slides, Fluoromount Aqueous Mounting Medium (Sigma) was applied, and slides were coverslipped. Images were taken with a 60× objective on a confocal fluorescent microscope Olympus FV 10i. and were processed using Olympus FluoView software (Ver.4.0a). Representative images from basolateral amygdala were taken from at least two sections from six-seven mice per group.

The numbers of immuno-positive cells/microscope field were quantified for all markers in the basolateral amygdala according to the Paxinos and Franklin stereotaxic atlas [80].

#### 4.6.2. Colocalization of Molecules

To analyze colocalization, the Mander’s overlap coefficient was assessed in some experiments. Counting was carried out in the field of view in the basolateral amygdala of the brain. The Mander’s overlap ratio indicates signal overlap, and thus reflects the true degree of molecular colocalization. The coefficient values range from 0 to 1.0. If the image has an overlap factor of 0.5, this means that 50% of the two investigated glows, i.e., pixels, overlap. A value of zero means that there are no overlapping pixels [81].

### 4.7. Insulin, Lactate and IL-1β Quantitative Analyzes

#### 4.7.1. Insulin Measurements

Insulin ELISA assays were done according to manufacturer’s instructions (Mouse Ultrasensitive Insulin ELISA 96 tests (80-INSMSU-E01)). For the study, amygdala homogenates prepared in phosphate buffered saline (PBS) were used. Since injections were carried out bilaterally, we took amygdala from both side for different measurements (six to seven mice per group). The sensitivity of the kit is 0.115 ng/mL. Protein concentration measurements were performed using the Bio-Rad protein assay kit and with bovine serum albumin as standards (Bio-Rad, Hercules, CA, USA). The resulting protein concentration was reported in mg/mL. The obtained values of the concentration of insulin in the sample in ng/mL were divided by the concentration of proteins in each individual sample. Thus, the result of the insulin evaluation in the homogenate was presented in ng/mg protein [82].

#### 4.7.2. Lactate Measurements

An enzymatic method followed by a colorimetric measure was used to analyze lactate concentration in amygdala, with a commercially available kit (L-Lactate Assay Kit, ref. ab65330, Abcam, Cambridge, UK) [83]. The kit sensitivity is >0.001 mM. Protein concentration measurements were performed using the Bio-Rad protein assay kit and with bovine serum albumin as standards (Bio-Rad, Hercules, CA, USA). The resulting protein concentrations were presented in μg/μL. The obtained values of the concentration of lactate in the sample in nmol/μL were divided by the concentration of proteins in each individual sample. Thus, the lactate concentration in the homogenate was presented in nmol/μg of protein.

#### 4.7.3. IL-1β Measurements

IL-1β concentration by enzyme-linked immunosorbent assay was carried out according to the protocol presented in the kit: IL-1 beta Mouse ELISA Kit 96 tests (KMC0011). For the study, amygdala homogenates prepared in a buffer containing 5M guanidine-HCl diluted in 50 mM Tris buffer, pH 8.0 and 1X PBS with 1X protease inhibitor were used. The measurement was carried out at 450 nm. The minimum detectable concentration of murine IL-1β was <7 pg/mL. Protein concentration measurements were performed using the Bio-Rad protein assay kit and with bovine serum albumin as standards (Bio-Rad, Hercules, CA, USA). The resulting protein concentration was reported in mg/mL. The obtained values of the concentration of interleukin 1β in the sample in pg/mL were divided by the concentration of proteins in each individual sample. Thus, the IL-1β concentration in the homogenate was presented in pg/mg protein.

### 4.8. Real-Time PCR

Total RNA was isolated from mouse brain subregions using RNA-Extran” (NPF Syntol, cat. No. EX-515) according to the manufacturer’s standard protocol. Reverse transcription was performed using the MMLV RT kit (Evrogen, cat. No. SK021) at 40 °C for 1 h. The resulting cDNA was used for real-time PCR using the qPCRmix-HS kit (Evrogen, cat. No. PK145L). We also used sets of primers and fluorescently labeled DNA probes to assess the number of cDNA fragments of the gene encoding insulin, as well as the reference genes GAPDH and ACTB (“DNA-Synthesis”, Russia). To determine the amount of insulin cDNA, as well as the reference ACTB and GAPDH genes, we used ready-made commercial reagent kits (produced by DNA-Synthesis, Moscow, Russia), which included two primers and a FAM-labeled probe. PCR was carried out using a LightCycler 96 (Roche) amplifier; relative quantitative analysis of gene expression was performed using the LightCycler 96 Software. Results are presented in arbitrary units (AU).

### 4.9. Statistical Analyses

Statistical analysis of the obtained results included descriptive statistics methods using the GraphPad Prism7 program (GraphPad Software, La Jolla, CA, USA). Within each sample, the arithmetic mean and the standard error of the mean were analyzed.

The Kolmogorov-Smirnov test was used to assess the normal distribution. The Mann-Whitney U test was used to compare differences between the two groups when it was not normally distributed. Repeated-measures two-way ANOVA followed by Tukey’s correction for multiple comparisons was used to assess the influence of NLRP3 knockout on Aβ-induced learning loss. Two-way ANOVA for independent samples with post hoc Sidak’s multiple comparisons was used when studying NLRP3 knockout contribution for Aβ-induced changes. The nonparametric Kruskal-Wallis test with Dunn’s post-hoc analysis was used when studying contextual and cued fear conditioning. A *p* value less than 0.05 was considered significant. All results were presented in the form of M ± SEM, where M is the average value, SEM is the standard error of the mean, *p* is the significance level.

## 5. Conclusions

Using an experimental approach to modeling Alzheimer’s disease, new molecular mechanisms of the dysregulation of insulin signaling in the amygdala in connection with the processes of neuroinflammation were investigated (Figure 12). NLRP3-dependent mechanisms were demonstrated in the amygdala in normal conditions and during the development of neurodegeneration. It was shown that NLRP3 inflammasomes are required for insulin-dependent glucose transport in the brain and memory consolidation. However, Nlrp3 knockout protected mice from the pathological effects of beta-amyloid oligomers and protected against the development of insulin resistance, which was manifested by the unchanged level of expression of IRS1-Ser compared to the control.

It was experimentally proven that preventing the development of local insulin resistance by blocking NLRP3 inflammasomes and mediated neuroinflammation should be considered a pathogenetically grounded approach to correcting cognitive disorders in Alzheimer’s disease. Taken together, our data revealed the protective role of Nlrp3 deletion in the regulation of fear memory and the development of Aβ-insulin resistance, providing a novel target for the clinical treatment of this disorder.

## Figures and Tables

**Figure 1 ijms-22-11588-f001:**
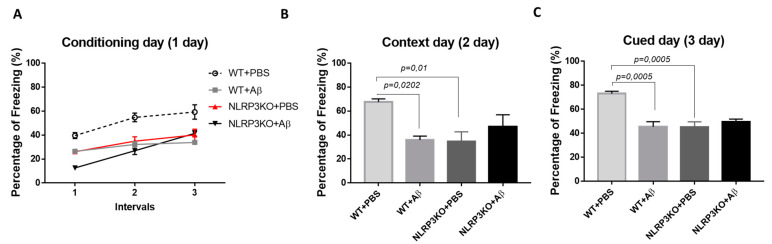
Results of behavioral testing in the Fear conditioning Test. Percentage of freezing on (**A**) the first conditioning day, (**B**) the second context day, (**C**) third signal day. WT—wild type control mice, NLRP3 KO—NLRP3 knockout mice, ns—not significant. Aβ—amyloid beta injection, PBS—phosphate buffered saline injection (sham operation).

**Figure 2 ijms-22-11588-f002:**
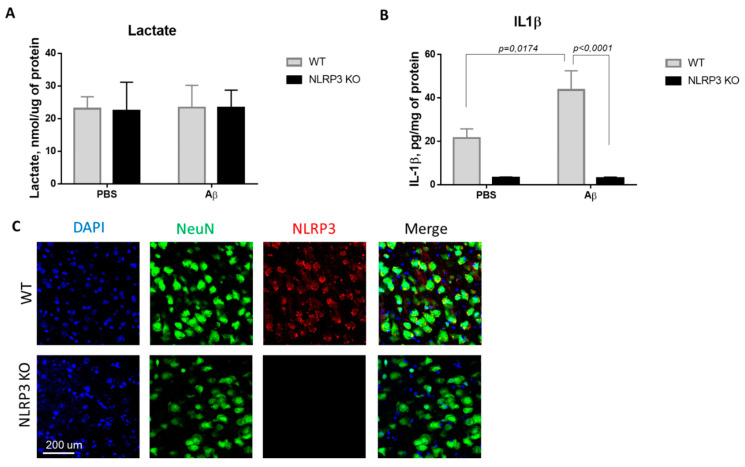
Results of measurements of (**A**) lactate in homogenates of the amygdala; (**B**) interleukin 1β in homogenates of the amygdala. (**C**) Representative confocal images of NLRP3-expressing neurons and stained for DAPI (blue), NeuN (green), NLRP3 (red), signal overlay (merge). Scale Bar 200 µm. NLRP3 KO^KO^—NLRP3 knockout mice, Aβ—amyloid beta injection, PBS—phosphate buffered saline injection (sham operation).

**Figure 3 ijms-22-11588-f003:**
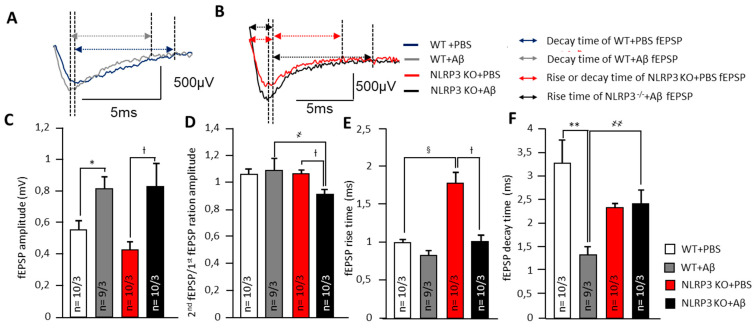
Alteration of synaptic transmission in synapses of amygdala neurons. (**A**) Representative traces of fEPSPs from WT animals marked with corresponding colors (blue—treated with PBS and grey treated with Aβ). Arrows shows the length of rise and decay time. (**B**) Representative traces of fEPSPs from NLRP3 KO animals marked with corresponding colors (red—treated with PBS and black treated with Aβ). Arrows shows the length of rise and decay time. The summary graphs show the average amplitudes (**C**)**,** PPF ratio (second amplitude/first amplitude fEPSP) (**D**), rise (**E**) and decay time (**F**) of fEPSPs. The groups are marked with corresponding colors. WT—wild type control mice, NLRP3 KO—NLRP3 knockout mice, Aβ—amyloid beta injection, PBS—phosphate buffered saline injection (sham operation). * *p* < 0.05; ** *p*< 0.01; † *p*< 0.05. ҂ *p* < 0.05; ҂҂ *p* < 0.01; § *p* < 0.05.

**Figure 4 ijms-22-11588-f004:**
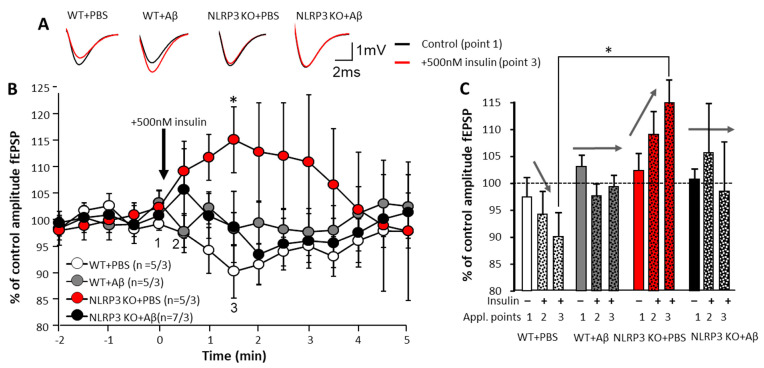
Effect of insulin application on synaptic transmission in the brain amygdala. (**A**) Upper panel shows representative traces of fEPSPs before (black) and after (red lines) 500 nM insulin application. (**B**) Lower panels show dynamics of pooled data for the fEPSPs amplitudes (control points 2 at 0.5 min and 3 at 1.5 min) normalized to the baseline period before insulin application (point 1 at time 0 min). The groups are marked with corresponding colors. (**C**) The summary graphs show the normalized average fEPSP amplitudes at points 1, 2 and 3 (See Figure 5B). Arrows shows the direction of fEPSPs amplitude dynamics in response to insulin application. WT—wild type control mice, NLRP3 KO—NLRP3 knockout mice, Aβ—amyloid beta injection, PBS—phosphate buffered saline injection (sham operation). * *p* < 0.05.

**Figure 5 ijms-22-11588-f005:**
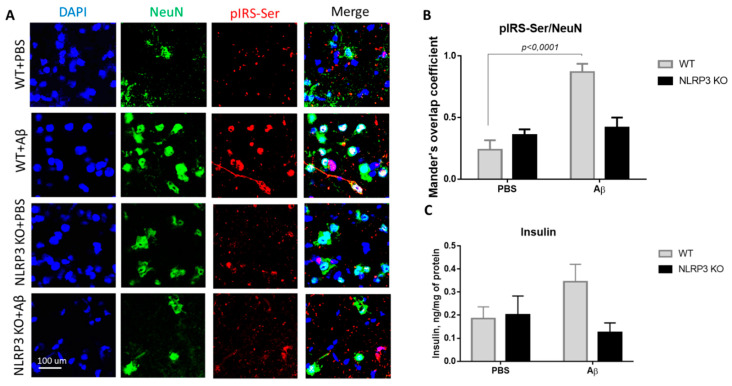
Results of insulin and serine phosphorylated insulin receptor substrate (pIRS-Ser) measurements. (**A**) Representative confocal images of pIRS-Ser-expressing neurons and stained for DAPI (blue), NeuN (green), pIRS-Ser (red), signal overlay (merge). Scale Bar 100 µm. (**B**) The Mander’s overlap coefficient in pIRS-Ser+ and NeuN+ double labeling cells. Data are presented as means ± SEM, two-way ANOVA. (**C**) Insulin measurement in homogenates of the amygdala (ng/mg of protein). WT—wild type control mice, NLRP3 KO—NLRP3 knockout mice, Aβ—amyloid beta injection, PBS—phosphate buffered saline injection (sham operation).

**Figure 6 ijms-22-11588-f006:**
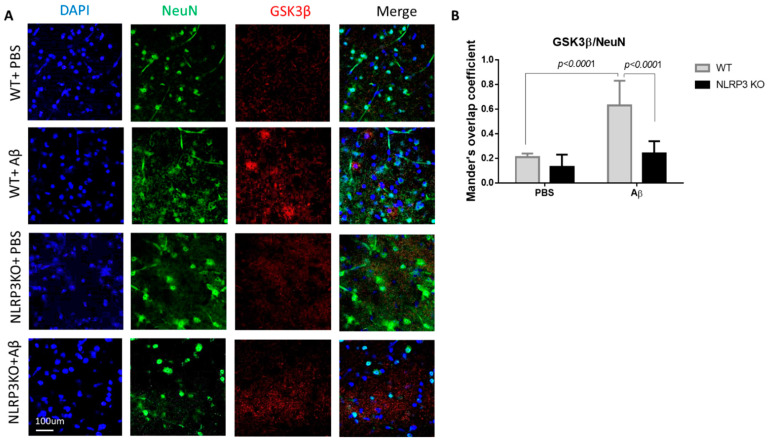
Results of glycogen synthase kinase (GSK3β) measurements. (**A**) Representative confocal images of GSK3β-expressing neurons and stained for DAPI (blue), NeuN (green), GSK3β (red), signal overlay (merge). Scale Bar 100 µm. (**B**) Mander’s overlap coefficient in GSK3β+ and NeuN+ double labeling cells. WT—wild type control mice, NLRP3 KO—NLRP3 knockout mice, Aβ—amyloid beta injection, PBS—phosphate buffered saline injection (sham operation). Data are presented as mean ± SEM, two-way ANOVA.

**Figure 7 ijms-22-11588-f007:**
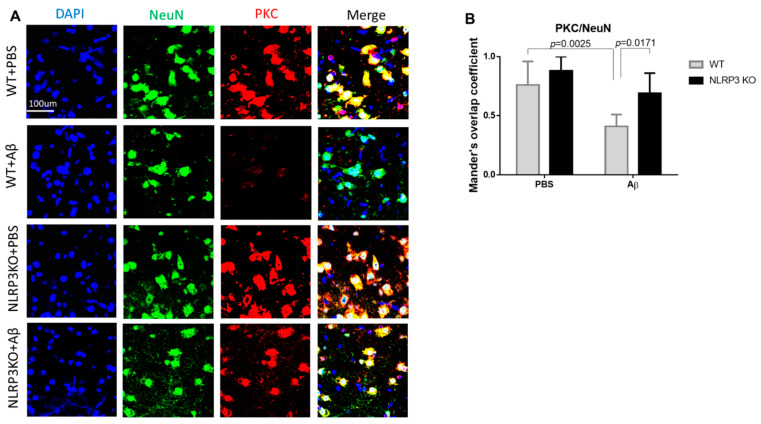
Protein kinase C (PKC) measurements. (**A**) Representative confocal images of PKC-expressing neurons and stained for DAPI (blue), NeuN (green), PKC (red), signal overlay (merge). Scale Bar 100 µm. (**B**) Mander’s overlap coefficient in PKC+ and NeuN+ double labeling cells. WT—wild type control mice, NLRP3 KO—NLRP3 knockout mice, Aβ—amyloid beta injection, PBS—phosphate buffered saline injection (sham operation). Data are presented as a mean ± SEM, two-way ANOVA.

**Figure 8 ijms-22-11588-f008:**
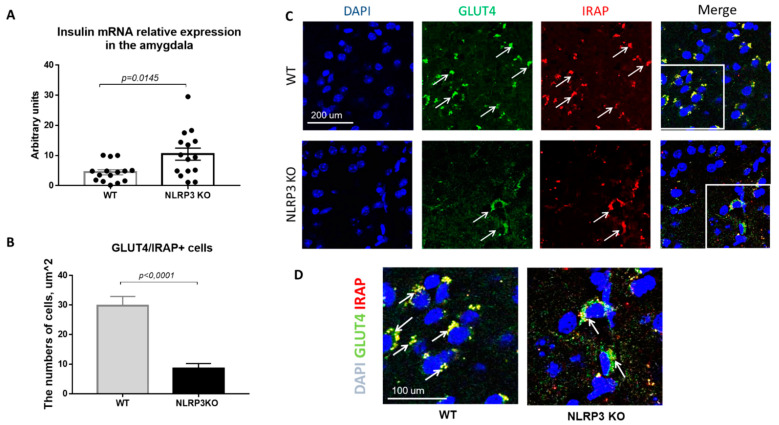
Evaluation of insulin mRNA and expression of GLUT4 and IRAP molecules in the basolateral amygdala of the mouse brain. (**A**) Insulin mRNA in arbitrary units (AU). (**B**) Number of GLUT4/IRAP + cells. (**C**) Triple immunofluorescent staining: cell nuclei—DAPI (blue), GLUT4 expression (green), IRAP expression (red), signal overlay (Merge). Scale Bar—200 µm. (**D**) Superposition of signals with triple immunofluorescent staining: cell nuclei DAPI (blue), GLUT4 (green), IRAP (red). Scale Bar—100 microns. WT—control mice group, NLRP3 KO—NLRP3 knockout mice.

**Figure 9 ijms-22-11588-f009:**
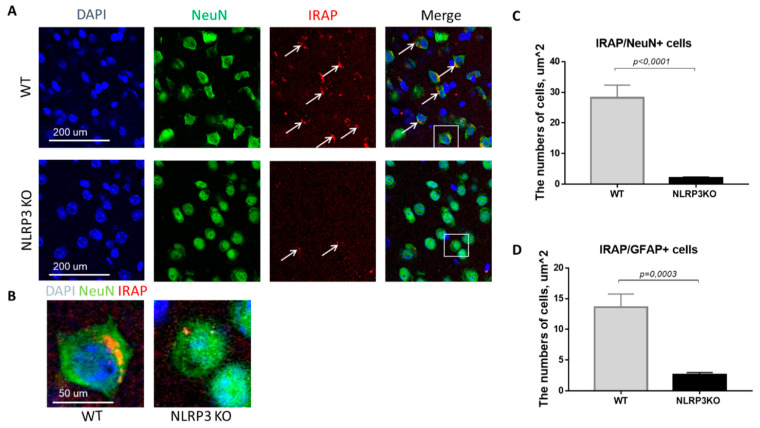
Expression of IRAP in mature neurons and astrocytes of the basolateral amygdala in mice. (**A**) Triple immunofluorescent staining: cell nuclei—DAPI (blue), NeuN (green), IRAP (red), signal overlay (Merge). Scale Bar—200 µm. (**B**) Overlay of signals: cell nuclei DAPI (blue), NeuN (green), IRAP (red). Scale Bar—100 µm. (**C**) Number of IRAP/NeuN + cells. (**D**) Number of IRAP/GFAP + cells. WT—control mice group, NLRP3KO—NLRP3 knockout mice. Data are presented as a mean ± SEM.

**Figure 10 ijms-22-11588-f010:**
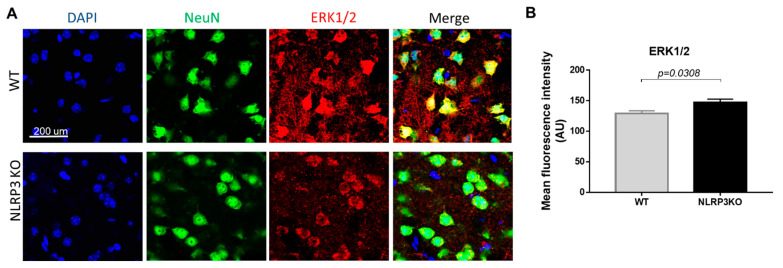
Expression of ERK1/2 in mature neurons of the basolateral amygdala in mice. (**A**) Triple immunofluorescent staining: cell nuclei—DAPI (blue), NeuN (green), ERK1/2 (red), signal overlay (Merge). Scale Bar—200 µm. (**B**) Mean fluorescence intensity of ERK1/2/NeuN + cells in arbitrary units. WT—control mice group, NLRP3KO—NLRP3 knockout mice. Data are presented as a mean ± SEM.

**Figure 11 ijms-22-11588-f011:**
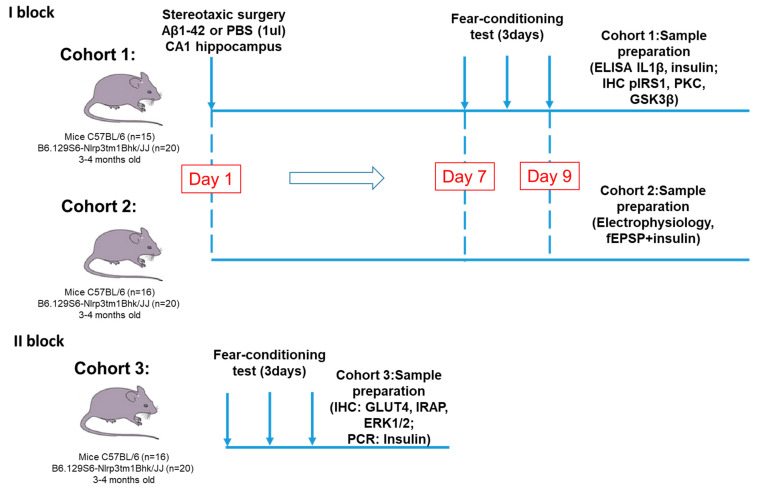
Experimental design. The study consisted of two main blocks. The first one was devoted to the investigation of the NLRP3 role in synaptic transmission, memory formation, glucose metabolism and insulin signaling in modeling Aβ-induced neurodegeneration. In the first block two cohorts of mice were used for immunohistochemistry and ELISA and for electrophysiological assessment. In the second block we evaluated the NLRP3 expression effect in the brain tissue on insulin-regulated transport of the glucose transporter. PBS—phosphate-saline buffer, Aβ—beta-amyloid, IHC—immunohistochemistry, PCR—polymerase chain reaction.

**Figure 12 ijms-22-11588-f012:**
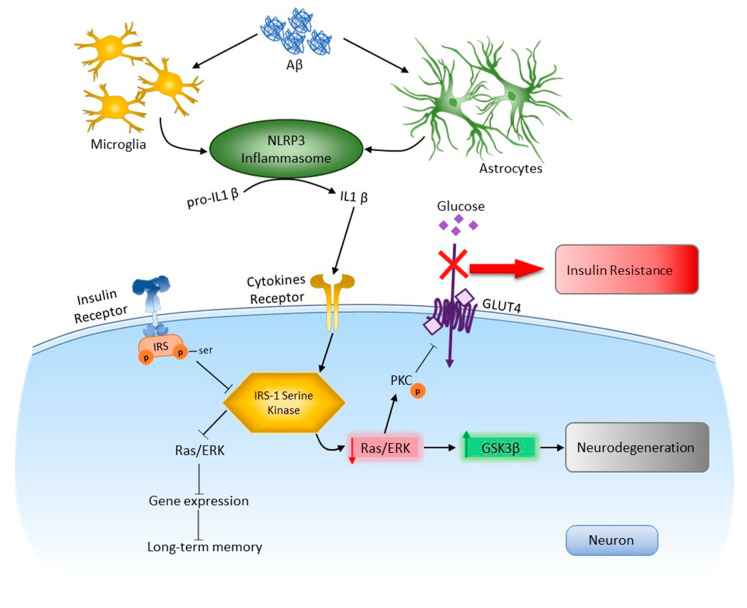
Role NLRP3 inflammasome in neurodegeneration and insulin resisance in response to DAMPs. In the brain, activation of toll-like receptors leads to the assembly of NLRP3 inflammasomes and the maturation and release of IL-1β. Through cytokine receptors, kinases are changed. This leads to impaired insulin signaling and the development of insulin resistance, which leads to dysfunction of long-term memory and the development of neurodegeneration. Aβ—amyloid beta, Akt—protein kinase B, GSK3β—glycogen synthase kinase 3 beta, IRS—insulin receptor substrate, NLRP3—NOD-like receptor pyrin domain-3 inflammasome, PKC—protein kinase C, Ras/ERK—kinase regulated by extracellular signals.

## Data Availability

The data presented in this study are available on request from the corresponding author.

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
