# Peer review of "NLRP3 Inflammasome Blocking as a Potential Treatment of Central Insulin Resistance in Early-Stage Alzheimer’s Disease"

_ijms, 2021, doi:10.3390/ijms222111588_

Round 1
Reviewer 1 Report
In the work by Komleva et al., titled "NLRP3 inflammasome blocking as a potential treatment of central insulin resistance in early-stage Alzheimer’s disease", the authors investigate the role of NLRP3 inflammasome in neurodegeneration in maintaining brain insulin signaling using both an induced model and KO mice, as well as several investigative techniques.
This original manuscript is well written and of great interest to the scientific community.
Line 27. Conclusion paragraph should be placed below.
Lines 39-40. “However, …unknown”. This sentence is not entirely true. Put at least one adequate reference to justify it.
Line 65. After “neurogenesis processes” excess space was left.
Line 74. “IRS-1” it should be put in full and not just the acronym.
Line 100. Change “understanding” to “understand”.
Figure 1, panel A. Why is the data represented separate?
The 4 groups should be represented all together, as the authors did in panels B and C.
The legend shows Ab, instead of Abeta, change it.
Furthermore, authors are invited to always call groups with the same name, that is: WT + PBS, WT + Abeta, NLRP3-KO + PBS and NLRP3-KO + Abeta. Keeping on changing the name of the groups makes reading and comprehension difficult (for example, lines 206, 221, etc “sham-operated control”).
Line 181. “Thus, …. After Abeta-injection” Report at what time it was performed.
Lines 197, 203 etc. Change “IL1β” to “IL-1β”.
Line 224. Why are the statistical written in this sentence (p<0.05) not shown in the figure?
Line 638. "(5-6)" What are you referring to? I do not think they are the mice per group, as the number is greater ... specify.
Line 638. Remove “animals”.
Lines 649-658. I advise the authors to reformulate this paragraph: above each division into groups, put the goal.
example:
The study consisted of two block:
I block: Investigation of the NLRP3 role in .... .
Animal were divided...
II block: Evaluation of the NLRP3 expression.... .
Animal were divided ...
Moreover, I advise authors, in block I to put first the name of the groups (bulleted list) and then the description.
Example:
- WT+PBS group. C5BL/6 mice with … etc
- WT+Aβ …
- … and so on.
Figure 11. The experimental scheme described by the authors does not best reflect Figure 11.
Blocks I and II should be better separated.
In the figure appear 2 cohorts, which are never described.
In the figure, do not all 4 groups appear and are the numbers of animals correct?
It can be assumed that the animals of each group are divided into two cohorts (half animals per cohort), to study what are different ... but it should be specified.
Also, were all animals (cohort 1 and 2) used for the behavioral test?
Line 673. Change “stereotactic surgery anesthesia” to “stereotactic surgery was performed in general anaesthesia with …”
Lines 684-688. If I understand correctly the mice were given 1 µl of Aβ or PBS, but then authors write that the animals were injected for 5 minutes, 1µl per minute. There is an inconsistency.
Were there any deaths during or after the procedure? How many?
Line 689. In addition to the fear conditioning test, were any other tests performed? With so many animals, just one behavioral test is a bit limiting ...
Line 699. Change “Basil” to “Basile”
Line 800. Change “An insulin ELISA” to “Insulin ELISA assay”.
Line 801. A second parenthesis must be closed after (80-INSMSU-E01).
Lines 798-829. The authors took the brain from the animals of cohort 1. Of some animals, the organ was fixed for immunohistochemistry, of some homogenized with PBS for insulin assay and of some homogenized in Tris-HCl for IL-1β assay. How were the animals (n) divided for these studies? It must be specified.
Furthermore, the lactate assay in which tissue was performed? It is not reported.
Line 830. Probe codes are missing.
General comments:
- Authors should review the entire manuscript in form. For example, “mg / ml” should be reported as “mg/ ml” (as always when "/" is present); or 37 ° C should be reported as 37°C. The unit of measurement should be placed seems or near, or spaced by the number, not a little and a little etc. Little things that need to be fixed.
- The authors mean the presence of neuroinflammation, but show only the cytokine IL-1beta. Authors are invited to show some other parameters.
Author Response
Dear Reviewer,
We wish to submit revised manuscript entitled “NLRP3 inflammasome blocking as a potential treatment of central insulin resistance in early-stage Alzheimer’s disease”.
We would like to thank the reviewer for providing constructive and thoughtful comments. Overall, the points raised were insightful and greatly appreciated. The revised manuscript has addressed the comments through clarification in the writing and/or presentation of the data, which we believe strengthen the overall work and conclusions of our findings. We hope these modifications to our manuscript are satisfactory for the reviewer.
Sincerely,
Yulia K. Komleva

Reviewer 2 Report
The authors have done interesting work that can add new information to the study of neurodegenerative diseases such as Alzheimer’s. However, some additions would increase the value of the article. For example, the authors in the introduction omitted the role of DAMPs and PAMPs in the genesis of neuroinflammation. Moreover, the authors could create a figure explaining the role played by immune cells in inflammatory genesis.
In section 2.6.1. Immunohistochemistry procedure. In line 770 lacks the antibodies dilution.
In addition, a more detailed explanation of the steps in the statistical analysis would be useful.
Authors should double-check the English language carefully and correct some typos and punctuation errors.
Regarding the bibliography there are several articles also very recent, so a remarkable good point for the manuscript.
Author Response

(The authors gave the same response as above.)

Round 2
Reviewer 1 Report
In this form, the manuscript can be accepted for publication